# Approaches for Modes of Action Study of Long Non-Coding RNAs: From Single Verification to Genome-Wide Determination

**DOI:** 10.3390/ijms24065562

**Published:** 2023-03-14

**Authors:** Xiaoyuan Tao, Sujuan Li, Guang Chen, Jian Wang, Shengchun Xu

**Affiliations:** 1Xianghu Laboratory, Hangzhou 311231, China; 2Central Laboratory, State Key Laboratory for Managing Biotic and Chemical Threats to the Quality and Safety of Agro-Products, Zhejiang Academy of Agricultural Sciences, Hangzhou 310021, China

**Keywords:** lncRNA, epigenetics, RNA–DNA interaction, RNA–protein interaction, ribonucleoprotein, chromatin accessibility

## Abstract

Long non-coding RNAs (lncRNAs) are transcripts longer than 200 nucleotides (nt) that are not translated into known functional proteins. This broad definition covers a large collection of transcripts with diverse genomic origins, biogenesis, and modes of action. Thus, it is very important to choose appropriate research methodologies when investigating lncRNAs with biological significance. Multiple reviews to date have summarized the mechanisms of lncRNA biogenesis, their localization, their functions in gene regulation at multiple levels, and also their potential applications. However, little has been reviewed on the leading strategies for lncRNA research. Here, we generalize a basic and systemic mind map for lncRNA research and discuss the mechanisms and the application scenarios of ‘up-to-date’ techniques as applied to molecular function studies of lncRNAs. Taking advantage of documented lncRNA research paradigms as examples, we aim to provide an overview of the developing techniques for elucidating lncRNA interactions with genomic DNA, proteins, and other RNAs. In the end, we propose the future direction and potential technological challenges of lncRNA studies, focusing on techniques and applications.

## 1. Introduction

The term long non-coding RNA refers to RNAs over 200 nt in length that do not encode known functional proteins [1,2]. Evidence over the past two decades indicates that lncRNAs are widely expressed in both animals and plants and play critical roles in gene regulation. The heterogeneity of lncRNAs in terms of genomic origin, biogenesis, and biological functions has been well discussed by multiple reviews [1,3,4,5]. Depending on biosynthesis, localization, and tissue- and condition-specific expression patterns, along with their specific interactions with DNA, RNA, or proteins, lncRNAs regulate function of chromatin [6,7], gene transcription [8,9], scaffolding/assembly and function of membraneless nuclear bodies [10,11,12], cytoplasmic mRNA stability and translation, and signaling pathways [13,14]. As such, lncRNAs are integral to gene expression regulation in myriad biological and physiopathological contexts such as neurodegeneration [15,16], immune responses [17], and cancer [18] in humans, as well as flowering [19], yield [20], and stress tolerance [21] in plants.

The Human GENCODE project annotated more than 16,000 lncRNA genes [22,23], but only a limited number of lncRNA transcripts have been experimentally confirmed as having biological functions. Are these RNAs merely by-products of transcription, or do they have some biological functions? The number and proportion of actually functional lncRNAs remain controversial. *Cis*-acting lncRNAs regulate gene expression in a manner dependent on the location of their own sites of transcription [20,24,25,26,27,28,29]. Meanwhile, *trans*-acting lncRNAs regulate gene expression at locations distant from its own transcription locus [12,30,31,32,33]. Both *cis* and *trans* regulation involve interactions of lncRNAs with specific DNA target sequences, other RNAs, or RNA-binding proteins (RBPs) including splicing factors, DNA modifiers, histone modifiers, and nucleosome remodeling factors.

In the past two decades, multiple approaches have been applied to elucidate the modes of action of biologically important lncRNAs. Taking advantage of the rapid development of next-generation sequencing, multi-omics methodologies have acquired an increasingly important role. Generally, ‘RNA-centric’ approaches investigate the possible interactions of DNA or proteins with known lncRNAs (e.g., RNA pull-down and chromatin isolation by RNA purification, ChIRP) [34], while ‘protein-centric’ approaches investigate lncRNAs that are possible interactors of known RBPs (e.g., crosslinking and immunoprecipitation, CLIP) [35]. These are mainly ‘one-to-one’ or ‘one-to-all’ approaches, which necessitate preliminary experiments to determine either a specific lncRNA with biological phenotypes or a specific RBP validated as having biological significance. Consequently, the detection throughput of these approaches is limited. More and more open-source ‘all-to-all’ methods have been applied in recent years to investigate the interactions of lncRNAs with RNA, DNA, and RBPs.

Multiple reviews to date have provided excellent discussion on the mechanisms of lncRNA biogenesis, their localization, their biological functions in gene regulation at multiple levels, and also their potential applications [1,5,36]. For comments on higher-resolution imaging and in vivo studies of lncRNAs, the readers are directed to excellent reviews and comments [37,38]. However, the progress and recently developed technological advances for lncRNAs have little been systemically summarized. In this review, the leading approaches for modes of action of lncRNA are summarized, and challenges and opportunities moving the field forward are discussed.

## 2. The Biochemical Mechanisms Underlying lncRNA Function

The functionality of lncRNAs is achieved through their interactions with other RNAs, DNAs, and RBPs. In interactions with DNA, the negative charge of RNA can neutralize the positively charged histone tail, causing chromatin to decompress [39]. Therefore, RNA-mediated chromatin opening and closing may play a role in rapid switching of gene expression. Mechanically, nuclear lncRNAs interact with DNA to change the chromatin environment in both *cis* and *trans* regulation, sometimes indirectly due to their affinity for proteins that can bind to RNA and DNA, and in other cases directly by binding DNA in a sequence-specific manner.

LncRNAs have the potential to form heterozygous structures with genomic DNA to affect chromatin accessibility. In one type of direct interaction, RNA can bind within the major groove of purine-rich DNA via Hoogsteen base pairing, forming a triple helical RNA–DNA structure that anchors the RNA to a specific DNA sequence and thereby targets RNA-associated regulatory proteins to distinct genomic sites. A growing number of examples have shown that the ability of lncRNAs to form RNA–DNA triplexes in vivo broadly affects chromatin accessibility [40]. The potential for RNA–DNA triplex formation depends largely on the RNA sequence, and there is already some machine learning software [41] that aims to enable the genome-wide prediction of such triplexes. Multiple lncRNAs are known to form RNA/DNA triplexes include *KHPS1* [42,43], *HOTAIR* [44], *PARTICLE* [44], *FENDRR* [45], and *GhDAN1* [46], indicating that formation of RNA–DNA triplex is a universal mode for lncRNA regulation of gene expression [47].

A more widely-studied pattern of lncRNA interactions with chromatin based on the principle of complementary base pairing (i.e., Watson–Crick base pairing) termed R-loop. R-loop has long been considered a threat to genomic stability. However, the transient properties of the R-loop also make it an ideal regulatory center, and recent studies have re-evaluated its functions in regulating gene expression and coordinating DNA repair [48,49]. With the help of proteins that recognize these structures, some lncRNAs regulate gene expression based on the R-loop, leading to a wide range of results. The development of the S9.6 antibody has made it possible to analyze R- loops through immunoprecipitation [50]. For example, DNA-RNA immunoprecipitation (DRIP) assays in mouse embryonic stem cells (mESCs) identified the lncRNA TARID (for TCF21 antisense RNA inducing demethylation) as forming an R-loop at the CpG-rich promoter of *TCF21* [51]. GADD45A protein recognizes and binds to the R-loop formed by the *TARID* lncRNA at the *TCF21* promoter to recruit the DNA demethylation factor TET1, resulting in transcriptional activation [52,53]. Both RNA/DNA triplexes and RNA/DNA R-loops act in a sequence-specific manner.

In addition to binding to sequence motifs, lncRNAs can also fold into secondary structures that interact with proteins involved in key signaling pathways [54]. As examples, *FAST* is transcribed in the antisense direction from the *FOXD3* gene and is highly expressed in hESCs, for which it is necessary to maintain pluripotency. Each *FAST* molecule forms five stem loops that provide a multivalent platform for the E3 ligase β-transducin repeats-containing protein (β-TrCP) and block the degradation of its substrate β-catenin, leading to activation of WNT signaling that is required for pluripotency [55]. The NF-κB interacting lncRNA *NKILA* forms two different hairpins, hairpin A and hairpin B, both of which bind to p65. Binding of hairpin B is necessary for *NKILA* to stably associate with the NF-kB:IkB complex to regulate NF-κB activity [56]. The secondary structure of *Xist* also implies an important relationship of RNA secondary structure with protein chaperone interaction. *Xist* contains six repeating sequence elements (called elements A-F) that mediate its interactions with respective protein partners such as ATRX, which binds *Xist* repeat A through its N-terminal RNA binding domain (RBD) and C-terminal helicase domain [57]. Moreover, repeats A and B account for two distinct phases of X-inactivation establishment [58]. Discrete *Xist* domains that interact with distinct sets of effector proteins have been revealed, such as DNMT1, CBX3, and phosphorylated CBX3 for the A domain; DNMT1, CBX3, pCBX3, and HNRNPH for the F domain; HNRNPU for the BCD domain; and CHD4, EZH2, SUZ12, DNMT1, and pCBX3 for the E domain [59]. The lncRNA *COOLAIR* at the *Arabidopsis FLC* locus [26,60] does not bind to PRC2 components, but rather mediates expression of *FLC* via alternative splicing [61,62]. Recently, a method based on single molecule sequencing (PacBio) to analyze the secondary structure of single RNA molecules in vivo, termed smStructure-seq, was applied to analyze the alternative splicing products of *COOLAIR*, revealing a hyper-variable region gives *COOLAIR* stronger chromatin binding ability, which leads to down-regulation of *FLC* expression [63]. Generally, accessibility is used to evaluate the exposure of each RNA base in vivo, as RNA–RNA pairing and RNA–DNA pairing lead to reduced accessibility. Such reduction is consistent with the findings in yeast and humans that lncRNA binding to chromatin can inhibit the transcriptional activity of genes. Taken together, the primary sequence of lncRNA is the basis of its interactions with RNA and DNA, and the secondary structure it forms plays an important role in interactions with RBPs. A series of in vitro and in vivo approaches have been developed to preserve and analyze these interactions.

## 3. Approaches for Study of lncRNA–Protein, lncRNA–DNA and lncRNA–RNA Interactions

### 3.1. Approaches for Study of lncRNA–Protein Interactions

Many lncRNAs play regulatory roles through recruitment of specific RNA–RBPs. Ramanathan et al. have systemically discussed the methods for studying RNA–protein interactions [64], in which the methods for detecting lncRNA–protein interactions were divided into two categories, including approaches characterize proteins bound to an lncRNA of interest (lncRNA centric), and approaches examine lncRNAs bound to a protein of interest (protein centric).

#### 3.1.1. lncRNA-Centric Approaches

RNA pull-down (Figure 1a) is the common “in vitro” procedure for characterizing proteins bound to an lncRNA of interest. First, the lncRNA of interest is labeled with 5′- or 3′- biotin or an aptamer tag (S1 or Cys4) through in vitro transcription. The labeled lncRNA is then incubated with cell lysate and functions as a ‘bait’ to bind interacting proteins. Next, the lncRNA–protein complex is isolated and purified based on the biotin-SA or S1 aptamer–resin system and subsequently assessed via Western blot (WB), protein microarray, or protein quantitative MS [64]. As an in vitro method, RNA pull-down might be affected by modifications to or the structure of the lncRNA and by post-translational modifications to and concentrations of proteins. A particular advantage of this method is the speed and ease of application for mutagenesis studies of a given RNA–protein pair. For example, in *Arabidopsis*, winter cold triggers enrichment of H3K27me3 in the chromatin of *FLOWERING LOCUS C* (*FLC*), a flower inhibitor, which leads to the epigenetic inhibition of *FLC*. This epigenetic regulation is mediated by the evolutionally conserved inhibitor complex polycomb inhibitor complex 2 (PRC2). The lncRNA *COLDWRAP* from *FLC* locus transcribed from the *FLC* promoter region in the same direction as *FLC* mRNA. RNA pull-down using biotin-labeled *COLDWRAP* has been shown to successfully enrich polycomb group CLF protein fused with his-tag [27]. Various “in vivo” lncRNA-centric approaches have also been developed, including RAP (RNA antisense purification) [65], PAIR (the peptide nucleic acid (PNA)-assisted identification of RBPs) [66], MS2-BioTRAP (MS2 In Vivo Biotinylated RAP) [67], TRIP (the tandem RNA isolation procedure) [68], CHART (capture hybridization analysis of RNA targets) [69], ChIRP (chromatin isolation by RNA purification)-MS ([34] (Figure 1b), and RaPID (RNA–protein interaction detection) [70]. They differ in the means of fixation, probe designation, and RNA–protein complex purification method. RAP technology uses long 120-nt 5′ biotinylated antisense probes to pull down RNA–RBP complexes after UV crosslinking and has been used to study noncoding RNAs such as *Xist* and *FIRRE* [71,72]. The PAIR technology uses a cell membrane–penetrating peptide (CPP) to deliver into cells a linked peptide nucleic acid (PNA) oligomer that complements the target RNA sequence. The PNA will then anneal to its target RNA, and covalently couples with it upon UV irradiation. The resulting PNA- RNA–RBP complex can be isolated using sense oligonucleotide magnetic beads, after which mass spectrometry (MS) is used to identify the RBPs [72]. MS2-BioTRAP technology takes advantage of the association between bacteriophage protein MS2 and RNA stem-loop tagging. In this method, HB (histidine-biotin)-tagged bacteriophage protein MS2 and stem-loop tagged target RNAs are co-expressed in cells. The tight association between MS2 and the RNA stem-loop tags allows efficient HB-tag based affinity purification of authentic RNA–protein complexes, which is subjected to MS identification [67]. TRIP technology has been applied to isolate in vivo crosslinked poly(A) RNA–protein complexes from cellular extracts with oligo(dT) beads, followed by the capture of specific RNAs with biotinylated antisense RNA oligonucleotides and streptavidin beads [68]. CHART and ChIRP enrich for lncRNA–protein complexes of interest using short biotinylated complementary DNA probes. CHART requires an additional RNase H step to identify accessible sites for oligonucleotide sequences, and sequences that lead to high RNase H sensitivity are used to design biotinylated probes [69]; meanwhile, ChIRP requires no prior knowledge of RNA accessibility for probe design, but rather applies a “split-probe” strategy, where all ~20-nt probes are split into two pools, “even” and “odd”, based on their relative positions along the target lncRNA. Two independent ChIRP-seq runs are performed, one each for the “even” and “odd” probes, which are enriched via purification with streptavidin (SA) magnetic beads. As the two sets of probes share no overlapping sequences, the only target they have in common is the RNA of interest and its associated chromatin [34]. RaPID technology uses an RNA component and a RaPID protein, in which the RNA component is comprised of bacteriophage lambda BoxB stem loops flanking any RNA motif of interest, and the RaPID protein component is comprised of a 22-amino-acid λN peptide fused to the N terminus of the HA-BirA* biotin ligase. The BoxB stem loops bind the λN peptide at high affinity and the stem-loop-tagged RNA recruits the RaPID protein, thereby biotinylating the proteins bound to the adjacent RNA motif of interest, which permits subsequent streptavidin capture, Western blotting, and mass spectrometry (MS) analysis. This proximity-dependent protein labeling can rapidly identify proteins that bind an RNA sequences of interest in living cells without cross-linking [70]. Given the diversity of methods available, the abundance of lncRNA and the strength of the studied lncRNA–protein interaction help to guide the selection of in vitro or in vivo approaches, and the UV- or formaldehyde-based cross-linking methods [64]. We searched PubMed using the keywords of “RNA antisense purification (RAP)”, “peptide nucleic acid (PNA)-assisted identification of RBPs”, “MS2-TRAP”, “tandem RNA isolation procedure”, “Capture hybridization analysis of RNA targets (CHART)”, “Chromatin isolation by RNA purification (ChIRP)” and “RNA-protein interaction detection (RaPID)”. These searches returned 22, four, five, two, six, 45, and two hits, respectively, with RAP and ChIRP so far being the top two lncRNA-centric protocols.

#### 3.1.2. Protein-Centric Approaches

Protein-centric approaches begin with an RBP of interest and aim to characterize its interacting lncRNAs. The most representative technologies are RNA immunoprecipitation (RIP) [73] (Figure 1c) and cross-linking immunoprecipitation (CLP) [74] (Figure 1d). Both RIP and CLIP use antibodies against the RBP of interest to pull down binding RNAs, subsequently followed with RT-qPCR for confirmation of a given lncRNA (‘one-to-one’ approach) or subjected to high-throughput sequencing [73] for ‘one-to-all’ screening of interacting lncRNAs. However, the two methods vary slightly in their principles and the details of their protocols. Unlike CLIP, RIP does not require UV cross-linking, which makes it easy to perform and guarantees repeatable and accurate results. For example, in *Arabidopsis*, *COLDAIR* is an intronic lncRNA with the same transcription direction as *FLC* and is necessary for *Arabidopsis* to properly inhibit *FLC* during vernalization [26]. In detecting this transcript, the polycomb group protein CLF was firstly fused with GFP, after which RIP was performed using a polyclonal antibody against GFP in *Arabidopsis* tissue during vernalization; *COLDAIR* transcripts were found in the immunoprecipitates after RIP [26]. In another example, it was found that *Xist* deletion triggers restoration of chromatin accessibility in some regions of X chromatin. ATAC-seq found that these peaks are significantly enriched for BRG1-related motifs, leading to speculation that BRG1 might play a role in determining the accessibility of Xi recovery areas. BRG1 RIP and subsequent qRT-PCR verified *Xist*, specifically the *Xist* repeats A, C, E, and F, to interact with BRG1. Similar methodologies were applied in the study of sno-lncRNA1-5 interact with alternative splicing regulator FOX2 [75], and also non-coding RNA *NORAD* interact with PUM proteins [76]. Various CLP-based methods including PAR-CLIP [77], iCLIP [78], eCLIP [35], irCLIP [79], GoldCLIP [80], fCLIP [81], and BrdU-CLIP [82] were developed. Ramanathan et al. have made a systematic comparison of the strengths and limitations of CLP family approaches [64]. To date, eCLIP [35] has probably produced the largest number of datasets as it was applied in the ENCODE project [64]. As an application of CLIP assay, *ANRIL* is an antisense transcript of the tumor suppressor *INK4b/ARF/INK4a* gene locus, which is inferred to be a polycomb factor-interacting lncRNA that directly affects expression of protein coding genes. CLIP assay of polycomb protein CBX7 and subsequent RT-qPCR successfully confirmed CBX7 binding of *ANRIL* transcripts [83,84]. The criteria for selection are as follows: if indirect lncRNA–protein interactions are tolerable and the binding sites within lncRNAs do not need to be determined, then the standard method is RIP. If either indirect interactions are not tolerable or RIP-seq is not satisfactory, CLP is the method of choice.

XRNAX (protein-crosslinked RNA extraction) (Figure 1e) and OOPS (orthogonal organic phase separation) are two methods for the generic purification of protein-crosslinked RNA [85,86,87]. Both XRNAX and OOPS use UV-crosslinking and conventional TRIZOL extraction [88], with the TRIZOL interphase being collected and washed to remove free proteins and RNA, followed by DNase-digestion to eliminate DNA and finally yield protein-crosslinked RNAs. XRNAX and OOPS are resourceful approaches that can be combined with various transcriptomic and proteomic methods including MS, IP-MS, and CLIP-seq to study the composition and dynamics of protein–RNA interactions [85,87].

Taken together, it is important to select optimal lncRNA–protein interaction approaches for the biological question being addressed. For confident identification of bona fide RNA–protein interactions, RBPs discovered via an lncRNA-centric method should be validated by the complementary protein-centric method, and vice versa, for lncRNAs discovered via an protein-centric method. Thus, to obtain robust supporting data, multiple approaches can be performed in parallel to support each other. In such endeavors, having an antibody with high efficiency is important; to date, available assay-verified antibodies include anti-H3K27me3 (Merck-Millipore), anti-EZH2 (active motif) [33], and anti-BRG1 (Abcam, ab110641) [89]. However, due to the expense and efficiency of generating a valid antibody, examples using protein-specific antibodies remain low in number compared with those expressing a tag-fused protein and employing a commercial anti-tag antibody (e.g., anti-FLAG).

### 3.2. Approaches for Study of lncRNA–DNA Interactions

The omics-based approaches for investigating RNA–DNA interactions detect the locations on the genome at which lncRNAs and their associated RNPs are bound; thus, the detections include both direct interactions of lncRNAs and indirect interactions through lncRNA-containing RNPs. Many lncRNAs play regulatory roles through targeting genomic DNAs. LncRNA-centric approaches used to elucidate lncRNA–protein interactions can also determine lncRNA–DNA interactions; for example, ChIRP followed by MS (ChIRP-MS) identifies RBPs that bind to an lncRNA of interest, while ChIRP followed with NGS sequencing (ChIRP-seq) identifies lncRNAs that target genomic DNA sites (Figure 1b). A number of resourceful approaches have been developed for ‘all-to-all’ screening of lncRNA–DNA interactions (Figure 2a), including MARGI (Mapping RNA-genome interactions) [90], CHAR-seq (Chromatin-Associated RNA sequencing) [91], GRID-seq (Global RNA interactions with DNA by deep sequencing) [92,93], and RADICL-seq (RNA and DNA Interacting Complexes Ligated and sequenced) [94]. These four methods are based on the ligation of RNA and DNA in proximity and share basic procedures such as cross-linking, chromatin/nuclei isolation, DNA fragmentation, proximity ligation, and library construction, although the detailed methodologies by which to achieve these procedures are method specific (Table 1). In general, cells and tissues are firstly cross-linked by formaldehyde (FA) or a combination of disuccinimidyl glutarate (DSG) and FA; secondly, a specific concentration of detergent is applied to lyse the cells and obtain nuclei or chromatin; thirdly, enzymes are used to digest the chromatin into DNA fragments. MARGI, CHAR-seq, and GRID-seq, respectively, use the restriction enzymes HaeIII, DpnII, and AluI to recognize specific sequences and cut the DNA at a specific location, while RADICL-seq applies DNase I to cut double-stranded DNA randomly at any site. In all cases, this digestion results in the release of RNA-proximal short fragments of DNA–RBP complexes. Fourthly, a nucleotide adaptor is ligated to the RNA and DNA in proximity, which is the key step of these technologies. The adaptors used vary among approaches (Figure 3), but have several characteristics in common; for example, all the adaptors are partially double-stranded with a 5′ single-strand overhang, and they are also pre-adenylated. In MARGI, GRID-seq, and RADICL-seq, the adaptors have a thymidine (T) overhang on the 3′ end for ligation with dsDNA fragments after end-repairing and dA tailing. Meanwhile, in CHAR-seq, the corresponding 3′ overhang of its adaptor is the DpnII complementary sequence (Figure 3); in addition, the adaptors all feature internal (T) biotinylation for subsequent purification in library construction. In all methods, the 5′ pre-adenylated single-strand overhang of the adaptor is ligated to the 3′ OH of RNA by ssRNA ligase, after which the 3′ double-strand end of the adaptor is ligated to the proximal DNA to form a RNA–DNA chimera structure. Finally, this chimera structure is reverse-transcribed to cDNA, which convert the information from the RNA sequences into DNA. Lastly, the molecules are purified using a biotin-SA system and NGS sequencing libraries constructed. Due to differences in adaptor sequences and enzyme restriction sites, the specific approaches for NGS library construction vary between methods; additionally, use of sonication or different restriction enzymes leads to differences in library fragment sizes. MARGI and CHAR-seq generate RNA–DNA libraries of random size; GRID-seq uses MmeI cutting, and thus, each RNA–DNA pair fragment is 20 bp; and RADICL-seq uses EcoP15I, yielding lengths of 27 bp. The fixed sequence length is convenient for subsequent bioinformatic analysis; however, it has the drawback of reduced unique mapping rate when sequencing a complex genome (especially that of a polyploid crop plant). Additionally, multiple strategies are applied to reduce invalid or meaningless ligation, including pre-adenylation of the 5′ overhang of the adaptor, which allows ligation of the 3′ end of the RNA in the absence of ATP; application of RNase H to digest the RNA/DNA duplex and reduce *cis* ligation with mRNA at the site of transcription; and, in RADICL-seq, the use of a hairpin adaptor to selectively ligate the bridge adapter that is covalently bound only to RNA and therefore prevent subsequent ligation of sequencing adapters. Ultimately, all four of these methods have confirmed their feasibility by validating previously reported lncRNAs. For examples, the lncRNA *MALAT1* has been detected by MARGI, GRID-seq, and RADICL-seq, and *roX2* can by GRID-seq and CHAR-seq. Additionally, GRID-seq was applied in *Arabidopsis* seedlings to identify more than 10,000 RNA–chromatin interactions mediated by protein-coding RNAs and non-coding RNAs, in which inter-chromosomal interactions were found to be primarily mediated by non-coding RNAs [51]. In addition, some RNA–chromatin interactions undergo alterations in response to biotic and abiotic stresses and form co-regulatory networks [51]. However, these approaches also have limitations. First, all require large amounts of tissue/cell input and hence obtain the average chromosome status of the cell population, which is not applicable for cells with high heterogeneity and small available input. Second, it takes at least 4–5 days to complete the complex protocols. Finally, a specialized pipeline that simplifies the downstream bioinformatic analysis is still lacking.

Two approaches, RedChIP (combining an RNA–DNA proximity ligation technique Red-C [100] with ChIP) [95] and ChRD-PET (chromatin-associated RNA–DNA interactions, followed by paired-end-tag sequencing) [96], have been developed to enable the identification of specific ribonucleoprotein (RNP)-related RNA–DNA interactions by combining proximity-based ligation of RNA and DNA and chromatin IP (Figure 2b). RedChIP employs the ligation first, then follows with specific RNP ChIP. Using the antibodies specific to the structural protein CTCF and the EZH2 subunit of polycomb repressive complex 2, RedChIP has been used to identify a spectrum of *cis*- and *trans*-acting ncRNAs enriched at polycomb- and CTCF-binding sites in human cells, which may be involved in polycomb-mediated gene repression and CTCF-dependent chromatin looping [95]. Meanwhile, ChRD-PET performs the ChIP first, then applies the proximity ligation to enriched IP complexes. This method has been used to develop a comprehensive interaction map between RNAs and H3K4me3-marked regions in rice, providing information on the H3K4me3 landscape and its related chromatin-interacting RNAs, both coding and non-coding, which engage extensively in the formation of chromatin loops and chromatin-interacting domains [96].

Recently, Quinodoz et al. developed RNA and DNA SPRITE to comprehensively map the spatial organization of RNA and DNA [97] (Figure 2c). The first iteration, SPRITE 1.0, predominantly detected RNA species with high abundance (e.g., 45S pre-rRNA) [97]; however, SPRITE 2.0 improved the efficiency of the RNA-tagging step and thereby enabled detection of all classes of RNA [98]. In the SPRITE process, DSG and FA are first used to establish crosslinks and preserve RNA and DNA interactions in situ. After lysis, the cells are sonicated briefly (1 min) and chromatin fragmented using DNase digestion to obtain lengths of approximately 150 bp–1 kb. The crosslinked complexes are immobilized on NHS (N-hydroxysuccinimide) magnetic beads which serve as a carrier in the several subsequent enzymatic steps; before this coupling, de-crosslinking and purification are performed and the concentration of RNA–DNA determined to estimate the needed amount of NHS beads. After DNA end repair and dA-tailing, five rounds of split-and-pool barcoding are applied in a 96-well plate in the following order: round (1) involves ligation of DNA Phosphate Modified (“DPM”) tags; round (2) involves ligation of the RNA Phosphate Modified (“RPM”) tags with high-concentration T4 RNA ligase I [101] after RNA overhang repair, followed by reverse transcription of the ligated RNA; and rounds (3–5) involve splitting of the beads across the 96-well plate in which each well contains a set of 96 different barcoding/index sequences for ligation, with round 3 being “Odd”, round 4 “Even”, and round 5 “Terminal” tags (Figure 2c) [97]. In this way, specific tags are ligated to the DNA molecules in each well, which leads to each crosslinked RNA–DNA molecule having a unique barcode sequence. Because all molecules in the cross-linked complexes are covalently linked, all members of a complex will be sorted into the same well and tagged with the same barcodes, while the molecules in different complexes will be split independently and so receive different barcodes. Moreover, the probability of molecules in two independent complexes receiving the same barcodes decreases exponentially with each additional round of split-and-pool barcoding; after six rounds, there may be 10^12^ possible unique barcode sequences, which exceeds the number of unique DNA molecules in the initial sample (10^9^ possibilities) [98]. Finally, the barcoded RNA–DNA fragments are de-crosslinked and purified using the biotin-SA system, after which library construction and NGS sequencing are performed. DNA or RNA sequences tagged with the same tag combination pattern are termed a SPRITE cluster. Notably, SPRITE is not limited to the paired interaction between two sites, but can detect multiple DNA and RNA molecules that interact at the same time and hence yields more information about the structure and function of chromatin interaction regions. Moreover, SPRITE can be used to detect DNA–DNA interactions (DNA SPRITE) and RNA–DNA interactions (RNA and DNA SPRITE), as well as RNA–RNA interactions. Thus, SPRITE enables genome-wide detection of multiple simultaneously occurring higher-order DNA and RNA interactions and elucidates how nuclear non-coding RNAs serve as spatial organizers controlling processes underpinning gene regulation. Application of SPRITE established hundreds of ncRNAs throughout the nucleus serving as spatial organizers to shape DNA contacts, heterochromatin, and gene expression [98]. There are three limitations of SPRITE: first, high sequencing depth was required to identify genomic architecture with high-resolution; second is the narrow window of chromatin fragmentation optimization to obtain meaningful contacts, and overfragmentation or underfragmentation affects the quality of libraries; third, there is also a high upfront cost to order the 96-well plate of adaptors with modifications [99].

### 3.3. Approaches for Study of lncRNA–RNA Interactions

Unlike protein-coding mRNAs, most lncRNAs are located in the nucleus, and many of them are associated with chromatin [102]. Accordingly, the approaches for mode of action study of lncRNA majorly focuses on lncRNA interactions with DNA and RBPs, and approaches developed for lncRNA–RNA interactions are relatively limited. LncRNAs are rich in microRNA (miRNA) complementary sites, so they can act as competitive endogenous RNAs or “sponges” of miRNAs, thereby reducing miRNA availability for targeting of mRNAs and regulation of gene expression [103,104]. The lncRNA–miRNA action pattern is well documented [103,104]. Traditionally, interactions between lncRNA and other RNA (e.g., miRNA, mRNA) can be determined by RNA pulldown followed by RT-PCR [105] or NGS (e.g., RIA-seq [13]). For example, in tumors, the *PNUTS* lncRNA is produced by alternative splicing of the *PNUTS* pre-mRNA. Bioinformatics predictions indicate that the *PNUTS* lncRNA contains seven binding sites for miR-205, and incubation of MS2 marker and biotin-labeled *PNUTS* lncRNA with cell lysates followed by miRNA-specific RT-PCR has shown that the *PNUTS* lncRNA can pull down miR-205. miR-205 is the mature inhibitor of the transcriptional inhibitors *ZEB1* and *ZEB2* and is required to maintain epithelial cells. Sponging of miR-205 by *PNUTS* leads to upregulation of *ZEB1* and *ZEB2*, thereby promoting epithelial–interstitial transformation and breast cancer cell migration and invasion [105]. Alternatively, the lncRNA was engineered with MS2-binding site repeats, and thus, the tagged lncRNA was immunoprecipitated by MS2-RIP, and its interacting RNA can be verified by RT-PCR [105].

RIC-seq (RNA in situ conformation sequencing) is the latest advance developed for the global profiling of RNA–RNA interactions [30,106] (Figure 2d). Generally, RNA in crosslinked nuclei are treated with micrococcal nuclease, the 3′ overhangs are dephosphorylated, the generated 3′ OH is labelled with a biotinylated cytidine (bis) phosphate (pCp–biotin), treatment with FastAP alkaline phosphatase is applied to remove the 3′ phosphate group from the Cp–biotin, and the RNA 5′ overhangs are phosphorylated with T4 polynucleotide kinase (PNK). After this, all resulting RNA fragments that are in close proximity are ligated in situ under non-denaturing conditions. The resulting chimeric RNAs are then enriched and strand-specific libraries constructed for sequencing. In all, RIC-seq is a powerful method for discovering the 3D structures, interactions, and regulatory roles of RNAs. Application of RIC-seq show that a super-enhancer hub RNA CCAT1-5L regulates MYC transcription via chromatin looping [30].

In many cases, the interaction of lncRNAs with DNA, RNA, or RNPs is not exclusive; there are many that interact directly or indirectly with DNA, RNA, and proteins at the same time. For example, *MEG3* is one of the lncRNAs that can be enriched in heterochromatin regions of breast cancer cells by RIP-seq using both anti-EZH2 and anti-H3K27me3 antibodies [33]. *MEG3* lncRNA is recruited to the chromatin of the target gene where its GA-rich RNA sequence and the GA-rich DNA sequence form an RNA–DNA triplex, and the PRC2 interacting region of *MEG3* promotes PRC2 recruitment to a distal regulatory element, thereby establishing H3K27me3 epigenetic modification to regulate gene expression [33]. For another example, the lncRNA *MALAT1* (metastasis-associated lung adenocarcinoma transcript 1) is probably the most abundant lncRNA in most cultured cells. *MALAT1* localizes to nuclear speckles, as indicated by RNA-FISH assay [31], fulfills important roles in pre-mRNA splicing and transcription, and is involved in cancer progression and metastasis. Application of the HyPR-MS approach [107], ChIRP-MS [108], and RNA pull-down-MS [107] has identified many *MALAT1* interacting proteins. In addition, application of RIC-seq has shown *MALAT1* to act as an RNA hub for many highly expressed RNAs. However, characterization using multi-color structured illumination microscopy indicated a nuclear speckle to be a multi-layered compartment at a higher resolution in which nuclear spot proteins such as the splicing factors SON and SC35 are localized in the center of the speckle, whereas *MALAT1* are is located on the periphery [12]. The unique pattern of *MALAT1* indicates it to promote nuclear speck formation and function. The PARIS method, which is based on reversible psoralen-crosslinking and applied for the global mapping of RNA duplexes with near base-pair resolution in living cells, identified *MALAT1* to form many long-range structures that may be involved in polyvalent interactions with different RBPs and pre-mRNAs [109]. Additionally, the RNA and DNA SPRITE assay indicated that hundreds of non-coding RNAs localize in spatial proximity to their transcriptional loci; however, some lncRNAs such as *MALAT1* localize broadly across all chromosomes [99]. Overall, the *MALAT1* lncRNA is a good model for studying lncRNA interactions with RNA and proteins.

### 3.4. Bioinformatic Processes for High-Throughput Approaches in Study of lncRNAs

Table 2 summarizes the bioinformatic processes for NGS data generated in lncRNA studies. Generally, processing of sequencing data includes three major steps: (1) read pre-processing, including filtering low-quality reads, adaptor trimming, and PCR duplicate removal; (2) alignment and post-alignment filtering; and (3) advanced analysis, data visualization, and interpretation. Some methods share similar data-processing workflows, for example, ATAC-seq, ChIP-seq, CUT&Tag, and ChIRP-seq all utilize similar alignment and peak calling processes but may differ in advanced analysis and data interpretation [34,110,111,112]. RNA–DNA proximity-ligation-based methods including CHAR-seq, GRID-seq, and RADICL-seq all need to separate the RNA and DNA pairs after read pre-processing, then map them separately to the genome to identify valid RNA–DNA interactions; however, the software used to perform the alignment may differ according to the sizes of the RNA–DNA pairs generated in these libraries [91,93,94]. Additionally, the tools used in advanced analysis and data interpretation vary greatly depending on the need to address the biological question. In particular, specific toolkits/pipelines have been developed for specific methodologies, such as RIPSeeker and ASPeak for RIP-seq [113,114], iMARGI-Docker for iMARGI [115], various tools including compartments caller, TAD callers, and interaction callers for chromosome conformation capture assay (3C)-based techniques [116], RedClib for RedChIP [100], and the SPRITE pipeline for SPRITE [99].

## 4. A General Mind Map for lncRNA Study

Given the diversity of available methodologies applied in lncRNA biology, we summarize a general mind map for lncRNA study (Figure 4). Traditional study of lncRNA include an understanding of ‘when’ a particular lncRNA is transcribed, ‘what’ phenotypes it related, ‘who’ it is, ‘where’ it localizes, and ‘how’ it acts [38]. First and foremost, it is necessary to determine the phenotypes related to an lncRNA and make sure that it has biological significance for the phenotype (‘what’ and ‘when’). This can be confirmed through joint analysis from multiple experiments, including physiopathological expression disorder [120], GWAS and map-based cloning [121], gain of function and loss of function mutants [122], overexpression or knockdown in animal cell lines, environmental responses and tissue specificity [123], function of the sense transcript corresponding to a natural antisense transcript (NAT), and alternative splicing [61]. Once the contribution of an lncRNA to gene expression and phenotype is confirmed, the underlying mode of its action can be further investigated. At this stage, the research methods usually employed with various pathological materials, transgenic plants, and transfected cell lines are RT-qPCR, non-coding RNA sequencing (especially long-read RNA-seq) [124,125,126,127], transcriptome sequencing, KEGG analysis, and other common detection and analysis methods, leading to a general understanding of whether lncRNA operates *in*-*cis* or *trans* (‘who’). It was further determined the sub-cellular and sub-organelle localization of lncRNA by single-molecule fluorescence in situ hybridization (smFISH) [128] combined with super-resolution imaging such as illumination microscopy (SIM) and stochastic optical reconstruction microscopy (STORM) [37,38,129], which is often indicative of the function it performs within the cell (‘where’). The information related to epigenetic modification by ChIP-seq, ATAC-seq or others which help to throw light on the versatile modes of action of lncRNAs, accordingly, proposed scientific hypotheses based on these results, which are further verified by biochemical approaches. The leading methodologies for mode of action study include three categories of lncRNA–DNA, lncRNA–protein, and lncRNA–RNA interactions (‘how’). Traditional robust methods can be used to study the molecular mechanisms of specific lncRNAs to confirm ‘one-to-one’ interactions between that lncRNA and its potential targets/partners. In recent years, a variety of ‘one-to-all’ approaches which combined with NGS sequencing or quantitative mass spectrometry (MS) have been applied in screening candidate RBPs for the lncRNA of interest, or candidate lncRNAs for the RBP of interest. Table 3 summarizes the application of methods in multiple well characterized lncRNAs. Finally, as the open-source methodology, the ‘all-to-all’ approaches provide the spatial organization of lncRNA and DNA, lncRNA and RNA, and lncRNA and protein interactions at a genome-wide scale along with the overall 3D structures of chromatin. Importantly, the candidate lncRNAs obtained by these ‘all-to-all’ methods need traditional phenotypic verification for further determination of their functions.

## 5. Conclusions and Perspective

Recent technical advances in methods for studying lncRNAs have shed light on complex and critical lncRNA–protein, lncRNA–DNA, and lncRNA–RNA interactions, which provided important profiles for lncRNA-mediated gene regulation. However, it is important for researchers to be aware of the strengths and limitations of different methods and to choose among them as best befits their research aims. Challenges to uncovering the biological functions of lncRNAs remain in many respects. Firstly, cutting-edge biochemical technologies will provide advanced sensitivity for versatile modes of action study of lncRNAs in examined biological pathways. It is still challenging to develop methodologies for the ‘all-to-all’ screening and profiling of lncRNAs with concise and precise performance and streamlined bioinformatic workflow. The operability, throughput, sensitivity, and reliability of such approaches still need validation and optimization. Secondly, it will be of great interest to determine lncRNA at the single-cell level, including its dynamic expression by single-cell sequencing technologies, and its localization (sub-cellular and sub-organelle) and dynamic and real-time binding with partner proteins by in vivo biochemical labeling followed by super-resolution imaging technologies. Additionally, single-molecule protein sequencing could provide single-molecule sensitivity detection for lncRNA interacting RBPs [153]. Thirdly, continuous development of more precise biochemical probing and labeling approaches for mapping the key modules of lncRNAs will provide new insights and expand our understanding of lncRNAs. For example, the discovery of RNA-specific Cas proteins such as Cas13 and Cas7-11 could be adapted to probe RNA–protein interactions in cells, or to label either RNA or protein at specific spots along an RNA by fusion of these Cas proteins with specific enzymes [154,155]. Fourthly, computational modeling in lncRNA folding and structure and the visualization of structure using cryoelectron microscopy (cryoEM) in real time are critical to systematically establish the lncRNA folding–function relationship. Finally, knowledge of lncRNAs has considerable promise in relation to potential therapeutic applications in humans and also in agricultural applications for crop improvement. In all, progress and advances in lncRNA methodologies will greatly facilitate our better understanding of the gene regulation modalities of lncRNAs.

## Figures and Tables

**Figure 1 ijms-24-05562-f001:**
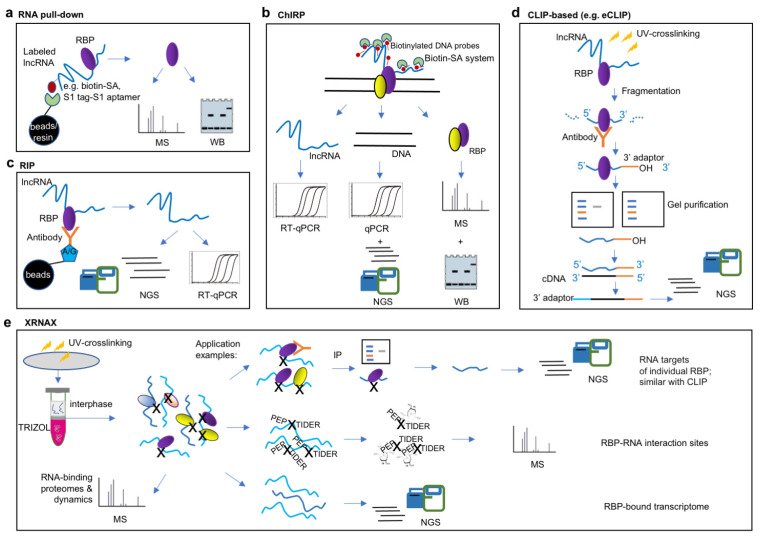
Methodologies for detecting lncRNA–protein interactions. (**a**) RNA pull-down is an “in vitro” procedure that use biotin or aptamer tag-labeled lncRNA as a ‘bait’ to bind interacting proteins, followed with identification of proteins by Western blot (WB) or protein quantitative mass spectrometry (MS) to characterize proteins bound to the lncRNA of interest. (**b**) Chromatin isolation by RNA purification (ChIRP) is based on affinity capture of target lncRNA:chromatin complex using biotinylated antisense probes of lncRNA, the resulting DNA and RBPs were determined by next-generation sequencing (NGS) and protein quantitative MS, respectively. (**c**,**d**) Both RNA immunoprecipitation (RIP) and crosslinking-immunprecipitation (CLIP)-based approaches use antibodies against the RBP of interest to pull-down its interacting RNAs. RIP was combined with RT-qPCR for confirmation of interaction with a candidate lncRNA, or subjected to NGS for ‘one-to-all’ screening of interacting lncRNAs (**c**). Specifically, CLIP-based approach, such as enhanced CLIP (eCLIP) uses UV cross-linking, fragmentation, immunprecipitation and gel purification to preserve the direct binding sites of target lncRNAs (**d**). (**e**) Protein-crosslinked RNA extraction (XRNAX) is a resourceful approach that purifies protein-crosslinked RNA of all biotypes from UV-crosslinked cells. XRNAX uses conventional TRIZOL extraction and collects the TRIZOL interphase for further purification, the resulting crosslinked protein-RNA complexes allows multiple applications including CLIP-seq, protein quantitative MS and NGS analysis for determination of RNA targets of individual RBP, RNA-binding proteomes and dynamics, RBP–RNA interaction sites and RBP-bound transcriptome.

**Figure 2 ijms-24-05562-f002:**
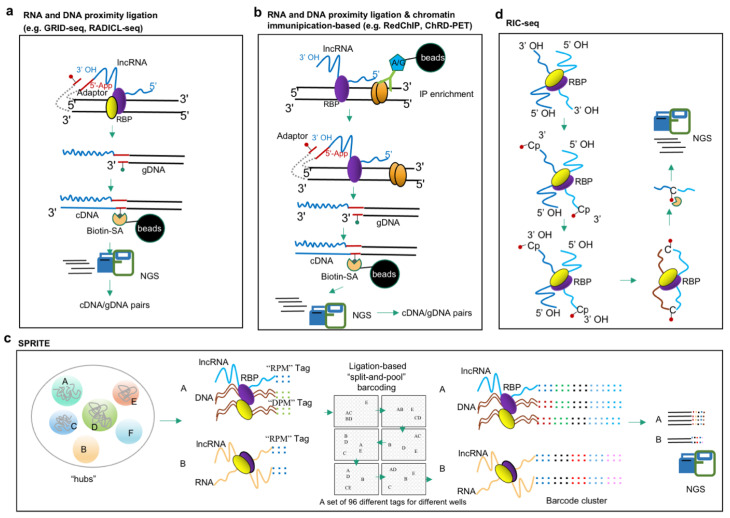
Methodologies for detecting lncRNA–DNA, and lncRNA–RNA interactions. (**a**) RNA and DNA proximity ligation-based approaches (e.g., GRID-seq, RADICL-seq) determines lncRNA–DNA interactions by in vivo proximity ligation of RNA and DNA using modified bridge linker (refer to Figure 3 for details), the resulting chimeric RNA–DNA were reversed transcribed and purified with biotin-SA system for NGS library construction and sequencing. (**b**) RNA and DNA proximity ligation and chromatin immunoprecipitation-based (e.g., RedChIP, ChRD-PET) combines RNA–DNA proximity ligation and chromatin immunoprecipitation (IP) enrichment for identifying RNA-chromatin interactions mediated by a particular protein or a particular histone modification. (**c**) SPRITE approaches map the spatial organization of RNA and DNA by repeatedly “split-and-pool” barcoding ligation over five rounds with a set of “RNA Phosphate Modified” (RPM), “DNA Phosphate Modified” (DPM), “Odd”, “Even” and “Terminal” tags. (**d**) RIC-seq globally captures protein-mediated RNA–RNA proximal interactions in vivo by labeling the 3′ end of interacting RNAs with pCp-biotin (biotinylated cytidine (bis) phosphate) and ligated to generate chimeric RNAs for NGS.

**Figure 3 ijms-24-05562-f003:**
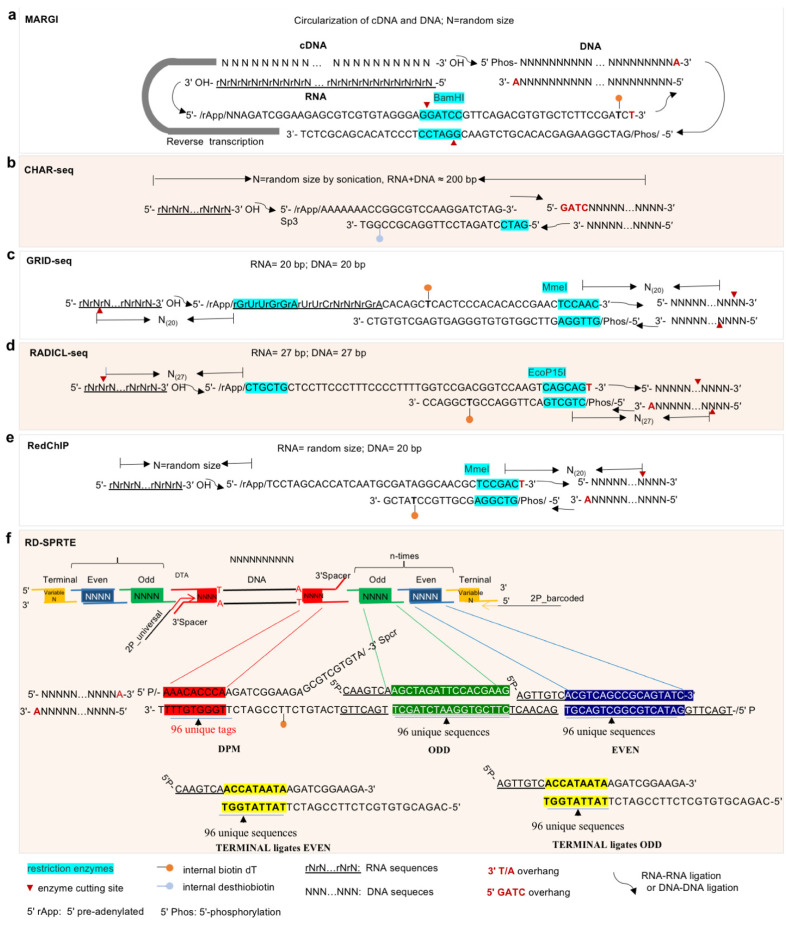
Schematic diagram showing the strategies for the design and modification of linker sequences in ligation-based approaches for identifying RNA–DNA interactions. (**a**) MARGI use double-stranded (ds) DNA linker with 10-nt 5′ overhang and 3′ A overhang at the top strand. The 5′ end of linker is adenylated (5′ rApp) for the ligation with the 3’ OH RNA. The 3′ end of dsDNA linker is phosphorylated (Phos) for proximity ligation with genomic DNA. Letters in blue: restriction enzyme site. N: random base of DNA. rN: random base of RNA. The lolypop icon indicates the internal biotin modification for further purification of products, the red triangle icon indicates the cutting site of restriction enzyme. (**b**) ChAR-seq use dsDNA linker with 6-nt 5′ overhang at the top strand and 4-nt 5′ overhang at the bottom strand. The 5′ end of linker is adenylated (5ʹ rApp) for the ligation with the 3’ OH RNA. The 3′ end of linker ligased to genomic DNA via sticky ends after *DpnII* digestion. (**c**) GRID-seq use a biotin-labeled bivalent linker consisting of a 14-nt single-stranded RNA (ssRNA) portion for ligation to RNA and a double-stranded DNA (dsDNA) portion for ligation to DNA. The linker was pre-adenylated at the 5′ end of the RNA and phosphorylated at the 3′ end of the dsDNA. Both 5′ end and 3′ end contain the *MmeI* restriction sites for NGS library construction, resulting as 20-bp sequence each for RNA and DNA in the ligased RNA–DNA pairs. (**d**,**e**) RADICL-seq and RedChIP use dsDNA linker with 5′ overhang and 3′ A overhang for ligation to RNA and genomic DNA, respectively. Differently, both ends of linker in RADICL-seq have *EcoP15I* restriction sites for preservation of 27-bp sequence each for RNA and DNA in the ligased RNA–DNA pairs, while the linker in RedChIP has *MmeI* restriction sites only in 3′ end for preservation of 20-bp sequence for the ligased genomic DNA. (**f**) RD-SPRITE use sets of DNA Phosphate Modified (“DPM” tags), “Odd,” “Even,” and “Terminal” tags for each round of “split-and-pool” ligation, each set of tags contains 96 different index sequences for barcoding.

**Figure 4 ijms-24-05562-f004:**
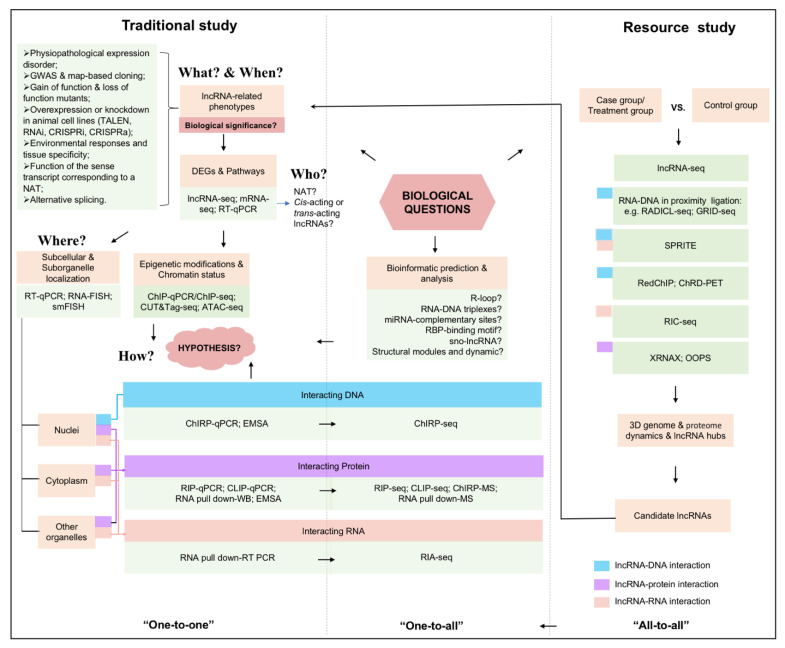
Schematic diagram summarizing the approaches used for lncRNA study. Traditional study of lncRNA include an understanding of ‘when’ a particular lncRNA is transcribed, ‘what’ phenotypes it related, ‘who’ it is, ‘where’ it localizes, and ‘how’ it acts [38]. The leading methodologies for mode of action study include three categories of lncRNA–DNA, lncRNA–protein and lncRNA–RNA interactions. ‘one-to-one’: approaches to verify single lncRNA–RBP interaction (usually followed by qPCR, RT-qPCR or WB), e.g., RIP-qPCR, RNA pulldown-WB; ‘one-to-all’: approaches to determine all the interacting RBPs of a specific lncRNAs, or all the interacting lncRNAs of a given RBP (usually followed by high throughput sequencing or MS), e.g., ChIRP-seq, CLIP-seq; ‘all-to-all’: resourceful approaches for genome-wide interaction profiling of lncRNAs with RNA, DNA, and RBPs, e.g., GRID-seq, SPRITE, and XRNAX.

**Table 1 ijms-24-05562-t001:** Comparison of ‘all-to all’ approaches for identifying lncRNA–DNA interactions.

	MARGI	CHAR-Seq	GRID-Seq	RADICL-Seq	RedChIP	ChRD-PET	SPRITE
Purpose	RNA–DNA interaction				RNA–DNA–protein interaction		RNA–DNA and DNA–DNA interactions
Mechanism	RNA and DNA proximity ligation				Combination of RNA and DNA proximity ligation and ChIP		Split-and-pool barcoding ligation
Applied organism/cells	Mammalian cells	Drosophila cells	Mammalian cells	Mammalian cells	Mammalian cells	Leaves of rice seedlings	Mammalian cells
Crosslinking	1% FA; or FA + DSG	1% FA	DSG + 3% FA	1% or 2% FA	1% FA	1% FA	DSG +3% FA
Nuclei isolation condition	NP-40 and SDS	Igepal, SDS, and Triton X-100	SDS	NP-40	NP40, SDS, and Triton X-100	Triton X-100 and SDS	Triton X-100 and NP-40
Chromatin fragmentation	Sonication or HaeIII	DpnII	AluI	DNase I	NlaIII	Sonication	Sonication and DNase
Bridge linker	See Figure 3						
Reduction in nascent transcription	N/A	N/A	N/A	RNase H	N/A	RNase H	N/A
Carrier between enzymatic steps	N/A	N/A	N/A	AMPure XP magnetic beads	N/A	Protein G magnetic beads	NHS-activated magnetic beads
Enzymes in ligation	T4 RNA Ligase 2, truncated KQ and T4 DNA ligase						T4 RNA Ligase 1 and T4 DNA ligase
Length of RNA and DNA pairs	Random size (~150 bp)	Random size (RNA + DNA ≈ 200 bp)	RNA = 20 bp; DNA = 20 bp	RNA = 27 bp; DNA = 27 bp	RNA: random size; DNA = 20 bp	Random size (~150 bp)	Random size (280 bp–1.3 kb)
Depth and sensitivity	105 million unique mapped read pairs corresponding to 2864 non-coding pxRNAs (proximal interaction) and 747 non-coding diRNAs (direct interaction)	22.2 million unique mapped read pairs corresponding to ~16,800 RNA transcripts	~40 million unique mapped read pairs corresponding to 868 mRNAs and 72 ncRNAs	~8.4 million unique mapped read pairs corresponding to 288,065 RNA–DNA interacting loci and 14,001 transcripts	18 ncRNAs specificly for CTCF and EZH2 proteins	~12.5 million unique mapped read pairs corresponding to 68,758 RNA–DNA interaction clusters	8 billion reads corresponding to 720 billion SPRITE clusters and ~650 lncRNAs
References	[90]	[91]	[92,93]	[94]	[95]	[96]	[97,98,99]

**Table 2 ijms-24-05562-t002:** Bioinformatic analysis for high-throughput methodologies in lncRNA studies.

Technology	Software/Code Used	References
CLIP or CLIP-based	Trimmomatic, TopHat, Piranha, PARalyzer, CLIPper, or Block-based peak calling	[117]
RNA immunoprecipitation (RIP); RIP-seq	RIPSeeker; ASPeak	[113,114]
Chromosome confromation capture assay (3C)-based techniques (3C-qPCR; ChIA-PET; Hi-C; Capture Hi-C)	Myriad tools including compartments caller, TAD callers, and interaction callers, along with visualization tools as summarized by Pal et al., 2019; CHiCANE toolkit	Capture Hi-C [118]; single cell Hi-C [116,119]
RIC-seq	FastQC, Trimmomatic, cutadapt, STAR; in-house script: remove_PCR_duplicates.pl, collect_pair_tags.pl, collect_pair_tags.pl, separate_intra_inter.pl, category_intra_reads.pl, cluster_intra_reads.pl, MonteCarlo_simulation.pl (GitHub: https://github.com/caochch/RICpipe; accessed on 21 May 2021)	[106]
ChIRP-seq	Bowtie, macs2	[34]
ChIP-seq; CUT&Tag; ATAC-seq	Fastp, hisat2, picard, macs2, deepTools, ChIPseeker	[110,111,112]
MARGI; iMARGI	iMARGI-Docker	[115]
CHAR-seq	FlyPipe (https://github.com/straightlab/flypipe; accessed on 20 February 2019), Super Deduper, Trimmomatic, bowtie2, SAMtools, BEDtools, MACS2, Circos, R, GraphPad Prism v7.0, deepTools2, HOMER, Geneious	[91]
GRID-seq	Cutadapt, bwa, samtools, GridTools.py, bgzip, tabix, Cytoscape	[93]
RADICL-seq	TagDust2, TagDust, RNAdust, FastUniq, BWA, samtools, bedtools, CAGEr package, ScoreMatrixBin package	[94]
RedChIP	RedClib (GitHub: https://github.com/agalitsyna/RedClib; accessed on 9 July 2020)	[100]
ChRD-PET	FastQC, Cutadapt, flash, BWA-MEM, BWA-ALN, HIAST2, BOWTIE2, BEDTools, HTSeq, MACS2, deepTools, plotBedpe function in the Sushi package in R, Seqtk, ggplot2 package, ggtern package	[96]
SPRITE	Conda, Snakemake, astq2json.py, config.yaml, Trim galore!, Cutadapt, Bowtie2, Bedtools, Multiqc, Samtools, Pigz, Fastqc, Python packages (Pysam, Numpy, R packages, Ggplot2, Gplots, Readr, Optparse); SPRITE pipeline (https://github.com/GuttmanLab/sprite-pipeline/wiki; accessed on 10 January 2022)	[99]

**Table 3 ijms-24-05562-t003:** Application of methodologies in specific lncRNA studies.

Technology	Aim	Example lncRNAs (Refs)	Recommended Protocols
smRNA FISH	Visualization and localization of lncRNA	*COOLAIR* [130]; *TINCR* [13]; *ANRIL* [83]; *LINC-PINT* [131]; *UMLILO* [132]	*Arabidopsis* [128]; Yeast [133]
CLIP or CLIP-based	Protein-centered method to identify specific RBP-associated lncRNAs	*ANRIL* [83]; *LINC-PINT* [131]	eCLIP [35]
RNA immunoprecipitation (RIP); RIP-seq	Protein-centered method to identify specific RBP-associated lncRNA	*DINO* [134]; *TINCR* [13]; *ANRIL* [83]; *UMLILO* [132]; *COLDAIR* [26]; *COLDWRAP* [27]; *LINC-PINT* [131]; *MEG3* [33]	[135]
DNA-RNA duplex immunopurification; DRIPc-seq	Immunopurification detection of R-loop	*APOLO* [136]	[137]
RNA pull-down	LncRNA-centered method to identify specific lncRNA-associated RNAs (pull down-PCR/pull down-seq) or proteins (pull down-WB/ pull down-MS)	*lncRNA-PNUTS* [105]; *TINCR* [13]; *ANRIL* [83]; *COLDWRAP* [27]; *LINC-PINT* [131]	[138]
ChIRP	Analysis of lncRNA chromatin targets (ChIRP-seq or ChIRP–qPCR) or protein interactors (ChIRP-MS)	*roX2* [34]; *TERC* [34]; *HOTAIR* [34]; *COOLAIR* [60]; *APOLO* [136]; *MEG3* [33]; *Xist* [7]; *DINO* [134]; *UMLILO* [132];	protocol with video [139]
Chromosome conformation capture assay (3C)-based techniques	3D genome architecture	*APOLO* [136]; *UMLILO* [132]	3C-qPCR [140]; ChIA-PET [141]; Hi-C [142]; Capture Hi-C [143]; Single cell Hi-C [144]
Electrophoretic mobility shift assay (EMSA)	In vitro detection of lncRNA–DNA triplex structures and lncRNA–protein binding	*MEG3* [33]; *ANRIL* [83]	[145]
ChIP	Immunopurification detection of DNA targets of lncRNA protein interactors (e.g., TFs and polycomb group proteins) and profiling of histone modifications	*COOLAIR* [60]; *APOLO* [136]; *MEG3* [33]; *ANRIL* [83]; *UMLILO* [132]; *LINC-PINT* [131]	Plant cells [146]; eChIP-Seq [147]; Animal cells [148]
CUT&Tag	Enzyme-tethered method to analyze DNA targets of proteins and histone modifications with low input	*MALAT1* [129]	Animal cells [149]; Plant cells [110]; scCUT&Tag [150]
ATAC-seq	Chromatin accessibility analysis	*DINO* [134]; *Xist* [151]	[152]

## Data Availability

Not applicable.

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
