# Peer review of "Approaches for Modes of Action Study of Long Non-Coding RNAs: From Single Verification to Genome-Wide Determination"

_ijms, 2023, doi:10.3390/ijms24065562_

Round 1

Reviewer 1 Report

The authors propose an excellent literature review on long noncoding RNAs, the manuscript is well structured and provides valuable and interesting indications.

A few minor improvements:

- figure 1 could be divided to make it more readable

- the part concerning miRNAs could be expanded, perhaps even adding a figure

- a paragraph on study limitations could be added

- the part relating to possible applications could be expanded, perhaps by summarizing them in a table

Reviewer 2 Report

Authors summarized the a basic and systemic mind map for lncRNA research, development technology and application of the research on the interaction between lncRNA and other molecules, and proposed the future direction and potential technical challenges of lncRNA research, which will be helpful to the scholars for studying lncRNA. However, the language of text writing is cumbersome, and the writing style is not conducive to readers' reading, which needs further major modification:

1. The introduction needs to be highly summarized.

2. It is suggested to merge the section 3 “Approvals for study of lncRNA-protein, lncRNA-DNA, and lncRNA-RNA inter-acts" with section 4 "application of technologies in lncRNA functional studies. For example, for each functional lncRNA (such as lncRNA-protein), the Approaches for study and their Examples of Application of technologies should be written together. 

3. Concussions and perspective also need to be condensed.

4. A small amount of grammar needs polishing so that readers can read the article smoothly.

Reviewer 3 Report

The review describes the approaches for detecting and investigating long non-coding RNAs.

Abstract may be revised to define long non-coding RNAs as "not translated into known functional proteins" etc. 

Table 2 may be revised to add more information on SPRITE.

Author Response

Response to Reviewer 3 Comments

Point 1: The review describes the approaches for detecting and investigating long non-coding RNAs.

Abstract may be revised to define long non-coding RNAs as "not translated into known functional proteins" etc. 

Response 1:

Thank you very much! We have revised the related sentences in the Abstract as well as in the Introduction part.

Point 2: Table 2 may be revised to add more information on SPRITE.

Response 2:

Very good suggestion!

We have added more information to Table 2 for the bioinformatic analysis of SPRITE.

We added:

“Conda, Snakemake, astq2json.py, config.yaml, Trim galore!, Cutadapt, Bowtie2, Bedtools, Multiqc, Samtools, Pigz, Fastqc, Python packages (Pysam, Numpy, R packages, Ggplot2, Gplots, Readr, Optparse); SPRITE pipeline (https://github.com/GuttmanLab/sprite-pipeline/wiki)”

Attached please find a marked version and a clean version of the  revised manuscript. The points in response to the other reviewers were also indicated. 

Thanks again.

Round 2

Reviewer 2 Report

The author has revised the article. I suggest that the manuscript can be published after checking the grammar in the clean version.